# Unmasking GluN1/GluN3A excitatory glycine NMDA receptors

Teddy Grand[1], Sarah Abi Gerges[2], Mélissa David[1], Marco A. Diana[2] & Pierre Paoletti[1]

GluN3A and GluN3B are glycine-binding subunits belonging to the NMDA receptor (NMDAR) family that can assemble with the GluN1 subunit to form unconventional receptors activated by glycine alone. Functional characterization of GluN1/GluN3 NMDARs has been difficult. Here, we uncover two modalities that have transformative properties on GluN1/GluN3A receptors. First, we identify a compound, CGP-78608, which greatly enhances GluN1/GluN3A responses, converting small and rapidly desensitizing currents into large and stable responses. Second, we show that an endogenous GluN3A disulfide bond endows GluN1/GluN3A receptors with distinct redox modulation, profoundly affecting agonist sensitivity and gating kinetics. Under reducing conditions, ambient glycine is sufficient to generate tonic receptor activation. Finally, using CGP-78608 on P8-P12 mouse hippocampal slices, we demonstrate that excitatory glycine GluN1/GluN3A NMDARs are functionally expressed in native neurons, at least in the juvenile brain. Our work opens new perspectives on the exploration of excitatory glycine receptors in brain function and development.

[1] Institut de Biologie de l'Ecole Normale Supérieure (IBENS), Ecole Normale Supérieure, Université PSL, CNRS, INSERM, F-75005 Paris, France. [2] Institut de Biologie Paris-Seine (IBPS) Sorbonne Université, CNRS, INSERM, Neurosciences Paris-Seine (NPS), UPMC Université Paris 06, F-75005 Paris, France. These authors contributed equally: Teddy Grand, Sarah Abi Gerges. These authors jointly supervised this work: Marco A. Diana, Pierre Paoletti. Correspondence and requests for materials should be addressed to M.A.D. (email: marco.diana@upmc.fr) or to P.P. (email: pierre.paoletti@ens.fr)

NMDARs are tetrameric ligand-gated ion channels that serve critical roles in CNS development and function. Normal NMDAR activity is essential for neuronal plasticity and information storage, while NMDAR dysfunction contributes to various CNS disorders including epilepsy, mental retardation, and schizophrenia[1–3]. Conventional NMDARs composed of two GluN1 and two GluN2 subunits require two agonists, glutamate and glycine (or D-serine), for activation[4]. They are highly permeable to $Ca^{2+}$, exhibit strong voltage-dependency due to $Mg^{2+}$ pore block, and cluster at excitatory synapses where they control synaptic strength by acting as coincidence detectors[5]. The functional diversity and signaling properties of GluN1/GluN2 NMDARs have been extensively characterized during the last 30 years, providing a wealth of information on the molecular basis of excitatory neurotransmission.

Much less is known regarding NMDARs incorporating the two glycine-binding subunits GluN3A and GluN3B, cloned in 1995 and 2001, respectively[6–10]. The general architecture of GluN3 subunits is globally similar to that of GluN1-2 subunits, yet the two families differ by the conspicuous presence of a positive charge in the pore-lining sequence of GluN3, and by the unique structural determinants of the GluN3 glycine-binding site[11,12]. GluN3A and GluN3B subunits also display differential ontogenic profiles. GluN3A is widely expressed in the CNS during early postnatal life and participates in synapse maturation[13–15], before progressively declining in abundance[16]. Conversely, GluN3B expression slowly increases throughout development, albeit remaining restricted to defined CNS regions (e.g. motoneurones[10,17]). GluN3-containing NMDARs are either glutamate/glycine-activated triheteromeric receptors composed of GluN1, GluN2, and GluN3 subunits or glycine-activated diheteromeric receptors composed of GluN1 and GluN3 subunits[18–20]. Both assemblies form cationic channels with strongly reduced $Ca^{2+}$ permeability and $Mg^{2+}$-block compared to GluN1/GluN2 receptors. Surprisingly, while the GluN3 subunit appears to act as a dominant negative regulator in triheteromeric GluN1/GluN2/GluN3 receptors[21,22], in GluN1/GluN3 diheteromers, the GluN3 subunit has a primary role in receptor activation[10,23,24].

Unconventional glycine-activated GluN1/GluN3 NMDARs have sparked intense curiosity and controversy. First, they constitute a new type of glycine excitatory receptor, since glycine is well established as an inhibitory neurotransmitter in the spinal cord and brainstem[25–27]. Second, they have so far been mostly described in recombinant systems with only sparse evidence for their existence in vivo[28]. Glycine currents produced by unconventional GluN1/GluN3 receptors are small, unstable and difficult to quantify[10,23,24,29–31], complicating the study of these receptors. Third, GluN1/GluN3 receptor pharmacology is meager. Antagonists with good potency and selectivity are missing, while available potentiators do not alter the transient nature of GluN1/GluN3 responses and usually produce complex biphasic effects[23,24,32–34]. In this study, we show that the quinoxalinedione CGP-78608, known to antagonize GluN1/GluN2 receptors[35], acts as an ultra-potent and powerful potentiator of GluN1/GluN3A receptors. The effects produced are unprecedented in their extent with potentiation factors much greater than previously observed with other GluN1-binding molecules. Building on this discovery, we also show that GluN1/GluN3A receptors can alternate between two modes of agonist sensitivity in a redox-dependent manner. In the high-affinity mode, ambient glycine is sufficient to tonically activate the receptors. Lastly, capitalizing on the discovery of the CGP-78608 as a massive enhancer of GluN1/GluN3A receptors, we identify excitatory glycine GluN1/GluN3A receptors as a new type of neuronal receptors, being functionally expressed in the juvenile brain. This work has broad ranging implications for the study of GluN1/GluN3A receptors and of glycine as an excitatory neurotransmitter.

## Results

### CGP-78608 awakens GluN1/GluN3A receptors.
Diheteromeric GluN1/GluN3 receptors display peculiar activation properties with no equivalent in the NMDAR family. In particular, contrasting with GluN1/GluN2 complexes, glutamate is dispensable for their activation[10]. Moreover, while glycine binding to GluN3 subunits triggers channel opening, glycine binding to the neighboring GluN1 subunits has an opposite effect causing auto-inhibition by rapid entry into a non-conducting desensitized state. Accordingly, receptors carrying single-point mutations that prevent glycine binding to GluN1 show large and non-desensitizing glycine-activated currents[23,24,30,33]. Taking advantage of these unique gating properties, we sought to find a small molecule compound able to antagonize glycine binding to GluN1 but not to GluN3A. Several GluN1-preferring ligands have already been shown to potentiate GluN1/GluN3 currents[23,24], yet none achieves high level of potentiation together with current stability during prolonged agonist application. Our attention focused on CGP-78608, a NMDAR GluN1 competitive antagonist[35] which, according to radiolabeled binding studies on isolated agonist-binding domains (ABDs), displays a thousand-fold selectivity for GluN1 vs GluN3[11]. We expressed diheteromeric GluN1/GluN3A receptors in HEK293 cells and studied their activity using whole-cell patch-clamp recordings coupled to a fast perfusion system. Application of glycine alone (100 μM) elicited very small (few tens of pA) and rapidly desensitizing currents (Fig. 1a), as previously described[30,31,34]. Pre-application of CGP-78608 (500 nM) dramatically enhanced the glycine-induced currents, which reached several nA in amplitude, and greatly reduced desensitization (Fig. 1a). Overall, peak and steady-state currents were potentiated $128 \pm 24$ fold (range 50–296; $n = 11$), and $335 \pm 266$ fold (range 106–876; $n = 11$), respectively (Fig. 1a). Concomitantly, the extent of desensitization was strongly decreased ($I_{ss}/I_{peak}$ increased from $0.30 \pm 0.03$ [$n = 8$] to $0.80 \pm 0.03$ [$n = 5$]; $P < 0.001$, Student's $t$-test) and desensitization kinetics slowed considerably ($\tau_{des}$ increased from $81 \pm 13$ ms [$n = 8$] to $1642 \pm 183$ ms [$n = 5$]; $P < 0.001$, Student's $t$-test). A full dose-response curve (Fig. 1b) revealed that CGP-78608 was extremely potent as a potentiator of GluN1/GluN3A-mediated glycine currents, with an estimated $EC_{50}$ in the low nM range ($26.3 \pm 5.0$ nM [$n = 4$–11]; measured at peak current), closely matching the value found on the isolated GluN1 ABD (6.4 nM; ref.[11]). In line with a classical bimolecular drug-receptor interaction, the effects of CGP-78608 were fully reversible (Supplementary Figure 1a). Furthermore, introduction of the GluN1-F484A mutation to prevent ligand binding to the GluN1 subunit[23,24] almost completely abolished the potentiating effects of CGP-78608 (Fig. 1d and Supplementary Figure 1b), consistent with CGP-78608 binding to GluN1 ABDs of GluN1/GluN3 receptors as it does on conventional GluN1/GluN2 receptors. Unsurprisingly, the amplitudes of glycine-triggered currents carried by GluN1-F484A/GluN3A receptors were much larger than those of wild-type GluN1/GluN3A receptors (Supplementary Figure 1b), confirming the inhibitory role on receptor activation of glycine binding to GluN1.

We obtained further evidence that CGP-78608 produces uniquely large potentiating effects on GluN1/GluN3A receptors by systematic comparison with other GluN1 competitive antagonists previously tested on these receptors[23,24,30]: MDL-29951, the kynurenic acid derivative 7-CKA, and L-689560. Similar to previous experiments, GluN1 competitive antagonists were first pre-incubated before triggering receptor activation by

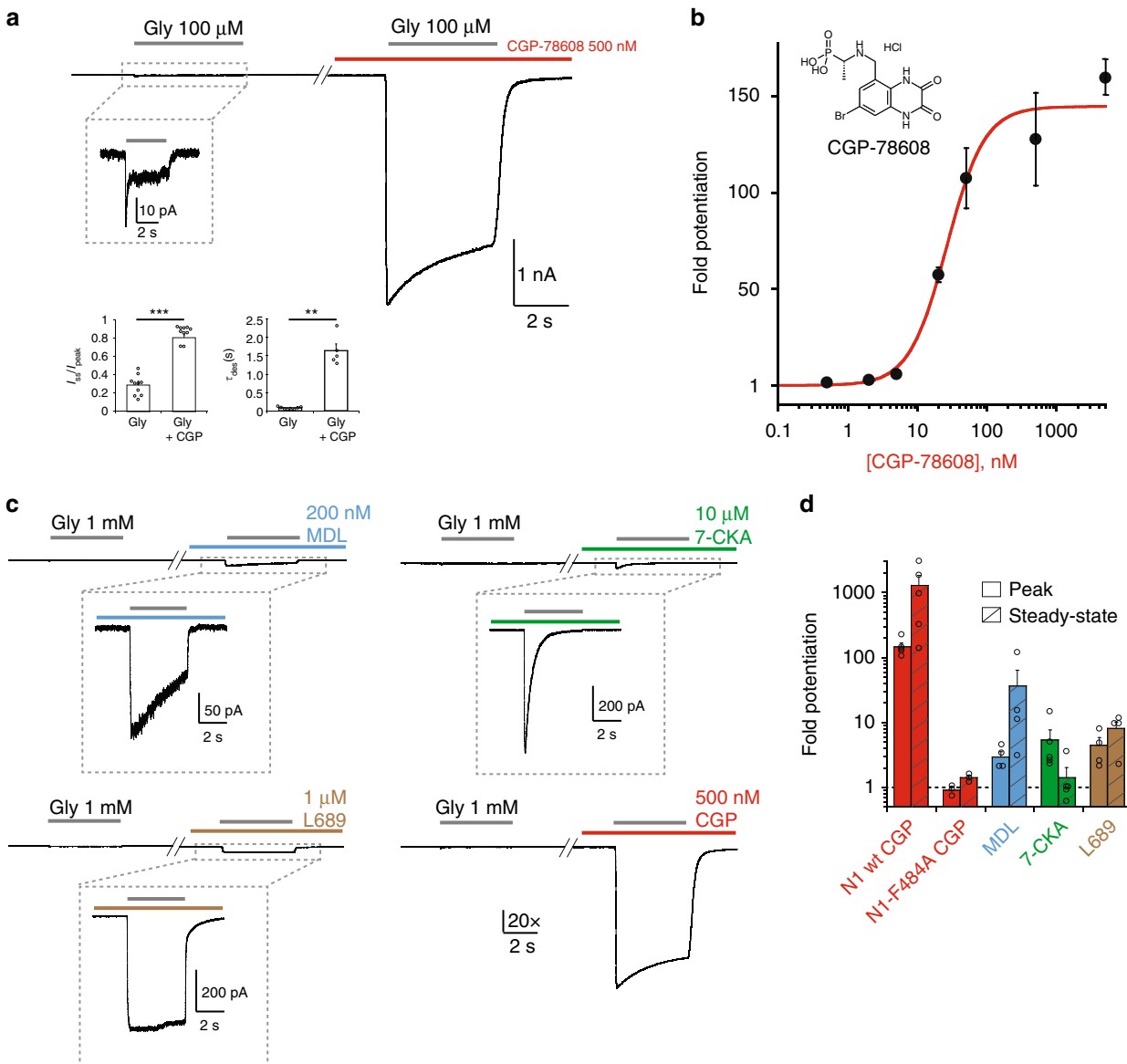

**Fig. 1** The GluN1 antagonist CGP-78608 awakes GluN1/GluN3A receptors. **a** Pre-application of CGP-78608 massively potentiates excitatory glycine GluN1/GluN3A receptor responses. Bar graphs indicate effect on extent ($I_{ss}/I_{peak}$) and kinetics ($\tau_{des}$) of desensitization. \*\*\*$P < 0.001$, Student's $t$-test, ($n = 10$); \*\*$P = 0.002$, Mann–Whitney test ($n = 10$). **b** CGP-78608 potentiation dose-response curve obtained in presence of 100 μM glycine. Currents were measured at the peak. $EC_{50} = 26.3 \pm 5$ nM, $n_{H} = 1.45$ ($n = 4$–11). **c** Representative current traces showing the relative potentiation of various GluN1 antagonists compared to CGP-78608. Traces are normalized to the glycine-induced peak currents obtained prior to drug application. **d** Quantification of effects of GluN1 antagonists tested in **c** on peak and steady-state current levels ($n = 4$-6). N1-F484A refers to the GluN1-F484A mutant subunit. All recordings were performed in HEK293 cells. Data are mean ± SEM

applying additional glycine (1 mM). The concentration of each compound was set above its reported GluN1 binding affinity (Ki; see Methods) to insure sufficient binding site occupancy. As illustrated in Fig. 1c, all three compounds potentiated GluN1/GluN3A receptors, yet their effects were singularly less prominent than those produced by CGP-78608. Thus, while CGP-78608 (500 nM) potentiated peak and steady-state currents by >100-fold ($n = 5$) and >1000-fold ($n = 5$), respectively, corresponding values were orders of magnitude lower for MDL-29951, 7-CKA, and L-689560 (Fig. 1d). Strikingly, for these three latter compounds, the magnitude of potentiation and the impact on the shape of the glycine currents differed significantly. 7-CKA produced the largest peak potentiation but desensitization remained profound with little current remaining at steady-state. In contrast, MDL-29951, and to a lesser extent L-689560, greatly

enhanced steady-state currents such that glycine responses were measurable during long agonist applications. These different patterns likely relate to the complex interplay between glycine and compound molecules for binding GluN1 and GluN3A subunits (see Discussion). Importantly, though, none of these three drugs produced effects comparable to CGP-78608, considering both the extent of peak and steady-state current potentiation. In agreement with its very high potency (i.e. low $EC_{50}$), washout experiments revealed slow offset kinetics of CGP-78608 from GluN1/GluN3A receptors, with dissociation time constants in the tens of seconds time scale (both from the active or resting state of the receptor; $\tau_{off}$ of $27.6 \pm 6.3$ s [$n = 6$] and $22.5 \pm 4.2$ s [$n = 4$], respectively; Fig. 2a). By inverting the order of application between CGP-78608 and glycine, we also found the CGP-78608 potentiation to be strongly state-dependent. Indeed,

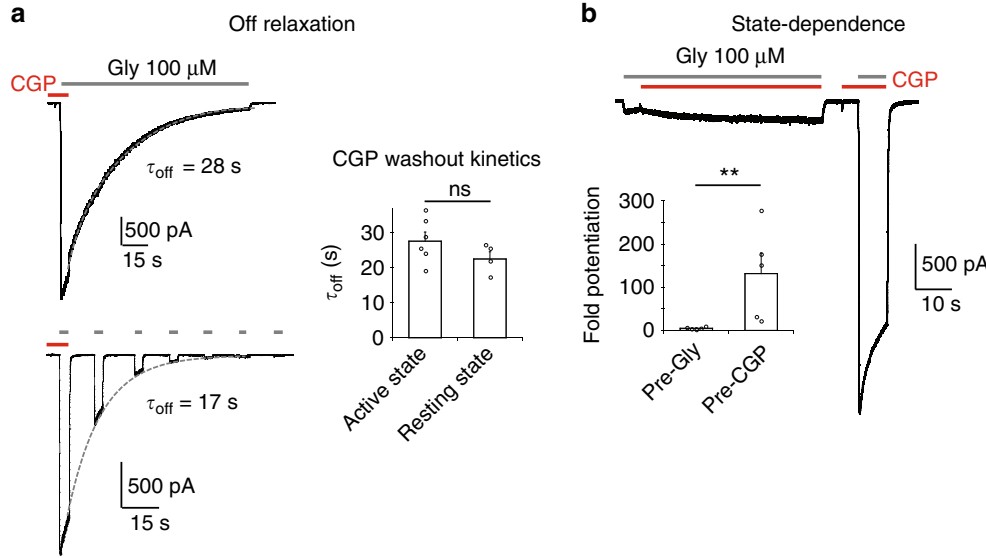

**Fig. 2** Kinetics and activity-dependence of CGP-78608 potentiation at GluN1/GluN3A receptors. **a** CGP-78608 washout kinetics in the presence of continuous or pulses of glycine. CGP-78608 was applied at 500 nM. Bar graph: comparison of the CGP-78608 washout kinetics in the active (upper trace) or resting state (lower trace). n.s. $P = 0.198$, Student's $t$-test; ($n = 4$-6). **b** CGP-78608 potentiation is state-dependent. Potentiation by CGP-78608 (500 nM) is greatly reduced when the compound is applied after, rather than before, application of glycine (pre-Gly vs post-Gly, respectively). Bar graph: potentiation measured at steady-state. **$P = 0.008$, Mann–Whitney test, ($n = 5$). Data are mean ± SEM

only minimal potentiation was observed on receptors pre-occupied with glycine, while CGP-78608 application before glycine still induced massive effects on the very same receptor population (Fig. 2b). This pre vs post pattern likely finds its origin in the relative affinities of GluN1 and GluN3 ABDs for glycine and CGP-78608 while the receptor transits along the activation pathway (from resting, to active and desensitized states; see Discussion). Using CGP-78608, we also confirmed that D-serine (up to 500 μM) triggered currents of smaller amplitude than those elicited by glycine (100 μM) (Supplementary Figure 1c), in agreement with D-serine having lower efficacy than glycine at GluN1/GluN3A receptors. Finally, we found all salient effects of CGP-78608 (current potentiation, reduction of desensitization, state-dependence) described in HEK293 cells to be conserved in *Xenopus* oocytes, highlighting the robustness of the CGP-78608 effects regardless of the expression system. In many oocytes, application of CGP-78608 unveiled massive glycine-induced currents (several μA), while prior applications of glycine alone on the very same cells virtually produced no detectable currents (Supplementary Figure 1d,e). These results provide additional proof that the compound CGP-78608 is unprecedented in its effects on excitatory glycine GluN1/GluN3A NMDARs. They also reveal that GluN1/GluN3A receptors, despite the small currents recorded when applying glycine only, express at very high levels in heterologous expression systems.

To better assess the activation parameters of GluN1/GluN3A receptors, we next performed glycine concentration-response curves in the presence or absence of CGP-78608. In control conditions, GluN1/GluN3A receptors displayed low μM glycine sensitivity ($EC_{50} = 7.1 \pm 0.4$ μM [$n = 5$–15], peak responses; Supplementary Figure 2a), in agreement with previous estimates[23,24]. In the presence of CGP-78608 (500 nM), in conditions where current responses are several orders of magnitude larger, glycine sensitivity was moderately decreased ($EC_{50} = 39 \pm 0.8$ μM [$n = 6$]; Supplementary Figure 2b). This ~5-fold rightward shift in agonist sensitivity may be due either to a direct competition between glycine and CGP-78608 binding to GluN3A ABD sites, or to an indirect (i.e. allosteric) effect of CGP-78608 binding at GluN1 ABD sites onto GluN3A ABD glycine

sites. To distinguish between these two possibilities, we repeated the concentration-response curve experiment with a 10-fold lower CGP-78608 concentration (50 nM). Under such condition, glycine $EC_{50}$ ($40 \pm 0.8$ μM; [$n = 9$]) is almost identical to that measured with 500 nM CGP-78608 (Supplementary Figure 2c). This result strongly suggests that CGP-78608 decreases glycine sensitivity of GluN1/GluN3A receptors through an inter-subunit allosteric effect between GluN1 and GluN3A ABD sites.

**Redox treatment transforms GluN1/GluN3A receptor properties.** Conventional GluN1/GluN2 NMDARs display a well-characterized redox sensitivity mediated by two critical GluN1 cysteine residues (C744 and C798) forming a disulfide bond, which increases the magnitude of NMDAR-evoked responses when chemically reduced[36–38]. Redox modulation is thought to provide an important regulatory mechanism of NMDAR signaling during normal and diseased brain function[39–43]. Because the redox sensitivity of GluN1/GluN3 receptors remains uncharacterized, we decided to investigate the effects of reducing treatments on glycine responses mediated by GluN1/GluN3A receptors. Recordings from HEK293 cells treated with the reducing agent TCEP (5 mM, 20 min) revealed drastic effects on GluN1/GluN3A glycine-evoked responses (Fig. 3b, c). The most conspicuous effect concerned the shape of the responses. First, when glycine was applied, peak currents from reduced GluN1/GluN3A receptors desensitized to a steady-state current that was more positive than the baseline holding current. Second, upon washout of glycine, very large and slowly decaying tail (or rebound) currents appeared. The size of the peak current induced by glycine application was also greatly increased following TCEP treatment. With the aim of identifying the molecular determinants underlying these effects, we first recorded from GluN1/GluN3A receptors lacking the endogenous GluN1 disulfide bridge responsible for the redox sensitivity of conventional GluN1/GluN2 NMDARs. Despite enhanced current amplitude prior to TCEP treatment, GluN1-C744S-C798S/GluN3A receptor currents remained remarkably sensitive to reduction, displaying the striking shape transformation observed on wild-type GluN1/

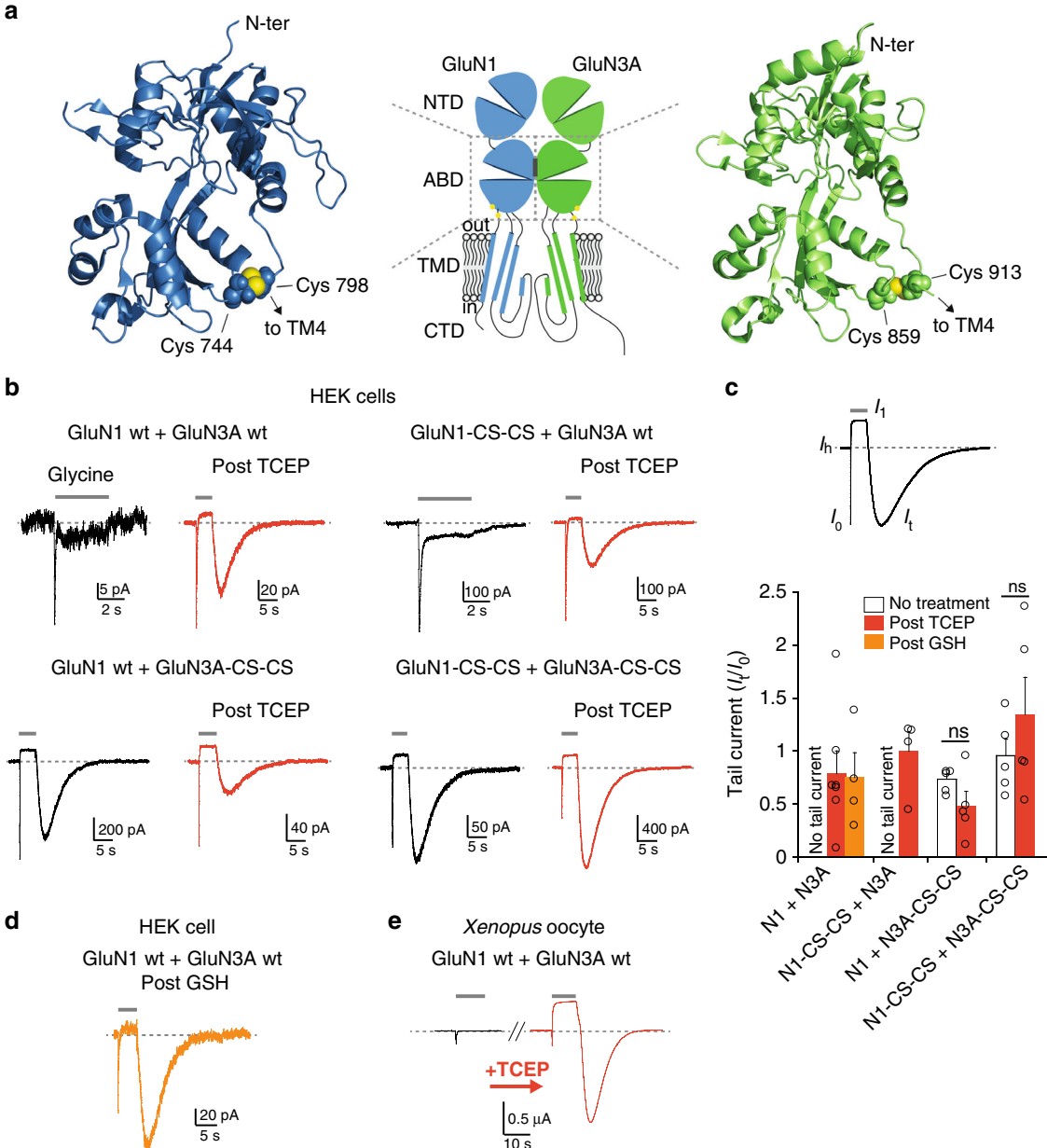

**Fig. 3** GluN1/GluN3A receptors display strong redox sensitivity. **a** Structure of the GluN1 and GluN3A ABDs (PDB 4KCC and 4KCD, respectively[55]). Cysteines involved in endogenous redox-sensitive disulfide bridges are highlighted (sulfur atoms shown in yellow). The central cartoon illustrates domain organization in GluN1/GluN3A receptors; NTD, N-terminal domain; ABD, agonist-binding domain; TMD, transmembrane domain; CTD, C-terminal domain. **b** Effect of TCEP treatment (5 mM, 20 min) on wild-type and mutant GluN1/GluN3A receptors expressed in HEK 293 cells. Glycine was applied at 100 μM. Each trace comes from a separate cell. **c** Quantification of tail currents observed upon washout of glycine on wild-type and mutant GluN1/GluN3A receptors. Inset indicates current tags used for quantification. n.s. $P = 0.125$ and $0.337$, Student's $t$-test ($n = 4$–$9$). Note that non-treated receptors containing wild-type GluN3A subunits do not exhibit tail currents. Data are mean ± SEM. **d** Effect of reduced glutathione (GSH, 50 mM for 20 min) on wild-type GluN1/GluN3A receptors expressed in HEK293 cells. Glycine was applied at 100 μM. **e** Effect of TCEP treatment (5 mM, 20 min) on wild-type GluN1/GluN3A receptors expressed in *Xenopus* oocytes. The pair of current traces corresponds to responses before and after TCEP treatment on the same cell. Glycine was applied at 100 μM

GluN3A receptors (Fig. 3b, c). A critical component of the redox sensitivity of GluN1/GluN3A receptors is therefore independent of the endogenous GluN1-C744-C798 disulfide bridge. Because a homologous disulfide bridge involving C859 and C913 is present in GluN3A[11] (Fig. 3a), we next recorded from mutant GluN1/GluN3A-C859S-C913S receptors. Distinct features emerged (Fig. 3b, c): first, the TCEP sensitivity (on current shape) was lost; second, the current responses in basal conditions (i.e. prior to TCEP treatment) were remarkably similar in shape to those of

reduced currents from wild-type receptors. Similar effects - absence of TCEP sensitivity, current shape transformation - were also present on receptors with the disulfide bonds mutated on both subunits (GluN1-C744S-C798S/GluN3A-C859S-C913S; Fig. 3b, c). Finally, we tested the endogenous reducing agent glutathione (GSH) on wild-type GluN1/GluN3A receptors. Similar to TCEP treatment, GSH incubation completely modified the current phenotype, with the appearance of an outward shifted steady-state current and large rebound currents upon removal of

glycine (Fig. 3c, d). Overall, these results show that GluN1/GluN3A receptors are highly redox sensitive, with activation properties that can adopt two strikingly different patterns depending on the redox state. They also identify the endogenous GluN3A ABD C859-C913 disulfide bridge as a key molecular determinant of this redox modulation.

We then turned to *Xenopus* oocytes to better estimate the relative contribution of the GluN1 and GluN3A subunits to the redox sensitivity of GluN1/GluN3A receptors. Indeed, we observed that oocytes better tolerated redox treatment than HEK293 cells, thus allowing robust comparison of currents from the same cell before and after reduction. When applying the reducing agent TCEP to wild-type GluN1/GluN3A receptors, currents radically changed, reproducing faithfully the effects previously observed on HEK293 cells. In particular, reduced receptors exhibited massive rebound currents upon washout of glycine, with much larger peak amplitudes than before TCEP treatment, and slow deactivation kinetics requiring tens of seconds for full recovery (Fig. 3e). Similar effects were observed when using the reducing agent DTE instead of TCEP (Supplementary Figure 3a, b). In addition, following the initial peak current triggered by glycine application, currents stabilized to a steady-state level lower (i.e. reduced inward current) than that measured prior to glycine application (Fig. 3e and see below). Having confirmed the reproducibility of the redox effects in both expression systems, we next aimed at deciphering the specific contribution of each subunit to the redox modulation of GluN1/GluN3A receptors. For that purpose, we assessed the redox sensitivity of receptors expressed in oocytes and lacking either the GluN1-C744-C798 disulfide bridge, or the GluN3A-C859-C913 disulfide bridge, or both. Comparison of mutant receptors revealed a clear dichotomy (Supplementary Figure 3a, b), where the GluN1 disulfide bridge controls current amplitude yet has little effect on the current shape, while the GluN3 disulfide bridge is responsible for the radical change in both current shape and time course.

**Tonic activation of reduced GluN1/GluN3A receptors.** Given the opposing effects of GluN1 and GluN3A subunits on GluN1/GluN3A receptor activation[23,24], we hypothesized that breaking the GluN3A-C859S-C913S disulfide bond shifts GluN3A glycine sensitivity, but not that of GluN1, to much higher affinity. Rebound currents upon glycine washout would then occur following fast dissociation of glycine from (inhibitory) GluN1 sites while (activating) GluN3 sites are still occupied. We obtained direct evidence that glycine dissociation kinetics are greatly slowed following GluN3A disulfide bond disruption by performing off-relaxation experiments on CGP-78608-bound non-desensitizing receptors. In such conditions, relaxation kinetics following glycine removal were readily observable and quantified (Fig. 4a, b). Comparison of wild-type and mutant receptors revealed that glycine off-relaxations ($\tau_{off}$) were only weakly affected when breaking the GluN1 disulfide bridge (442 ± 208 ms [$n = 5$] for GluN1-C744-C798/GluN3A receptors vs 205 ± 110 ms [$n = 7$] for wt receptors, $P = 0.027$, Student's *t*-test; Fig. 4b). In contrast, major effects were observed on receptors lacking the GluN3A bridge with glycine deactivation kinetics greatly slowed down (2217 ± 91 ms [$n = 4$] and 2098 ± 196 ms [$n = 5$] for GluN1/GluN3A-C859S-C913S and GluN1-C744S-C798S/GluN3A-C859S-C913S receptors, respectively; $P < 0.001$, Student's *t*-test comparison with wild-type or GluN1-C744-C798/GluN3A receptors; Fig. 4b). Interestingly, similarly slow time courses were measured for the decay kinetics of rebound currents (Supplementary Figure 3a, b), likely reflecting slow dissociation of glycine from GluN3A (see Discussion). Full glycine

concentration-response curves performed in the presence of CGP-78608 confirmed that mutant receptors displayed enhanced agonist sensitivity compared to their wild-type counterparts (13-fold decrease in glycine EC$_{50}$; Supplementary Figure 2d). Taken together, these results show that the endogenous GluN3A-C859-C913 disulfide bridge plays a major role in controlling glycine sensitivity of GluN1/GluN3A receptors. It does so by governing the residency time of glycine on its GluN3A ABD binding site.

We hypothesized that another consequence of the greatly enhanced glycine sensitivity of reduced GluN1/GluN3A receptors are outward shifted steady-state currents (see Fig. 3 and Supplementary Figure 3). The observation that agonist-induced steady-state currents are smaller than resting currents (measured prior to agonist application) is very unusual and may originate from combined effects of receptor desensitization and tonic activation. Accordingly, prior to glycine application, a fraction of reduced GluN1/GluN3 receptors would be constitutively activated by ambient glycine (estimated at 40–50 nM in recording solutions[44–46]). Following application of high glycine concentrations, more receptors activate, accounting for the fast inward peak current. Subsequently, however, desensitization would take over progressively accumulating all (or the vast majority of) receptors into a long-lasting inactive state, hence accounting for the apparent outward shift in steady-state current level. As illustrated in Fig. 4c, outward current shifts (represented as down bars) were directly and specifically controlled by the redox state of the GluN3A-C859-C913 disulfide bridge, paralleling the effects seen on the rebound currents. To better identify the nature of the constitutive currents recorded in the absence of added glycine, we used CNQX, an antagonist of GluN1/GluN2 NMDARs and AMPA receptors, also known to inhibit GluN1/GluN3A receptors[23,28]. For cells expressing wild-type receptors, CNQX (50 μM) barely changed the holding current (outward current displacement of 0.6 ± 0.06 pA, [$n = 6$]; Fig. 4d), as expected if the receptors were not tonically active. In contrast, for cells expressing GluN1-CS-CS/GluN3A-CS-CS receptors, CNQX strongly affected the holding current, displacing its amplitude by 22.9 ± 8.3 pA, [$n = 7$] ($P = 0.032$ compared to WT, Student's *t*-test), thus indicating a significant level of constitutive receptor activation (Fig. 4d). As expected, CNQX also inhibited the large tail currents carried by mutant receptors following glycine removal (Fig. 4e). Altogether, these results indicate that according to their redox state, GluN1/GluN3A receptors can switch mode, with differential glycine sensitivity. A single endogenous disulfide bond, residing in the GluN3A ABD lower lobe (GluN3A-C859-C913), controls this functional switch. In receptors with this S–S bond unlocked, glycine concentrations as low as a few tens of nanomolar appear sufficient to permit GluN3A subunit occupation and receptor activation. This ultra-high glycine sensitivity is unprecedented for functional GluN1/GluN3A receptors. However, it echoes surprisingly well the 40.4 nM glycine affinity found by Yao and Mayer[11] using radiolabeled ligand assay on the isolated GluN3A ABD.

**Functional GluN1/GluN3A receptors in the juvenile brain.** The existence of functional glycine excitatory GluN1/GluN3A NMDARs has not been demonstrated in native neurons until present. In native conditions, similar to heterologous systems, glycine binding to GluN1 subunits may reduce currents to undetectable levels because of receptor desensitization. Given its properties on recombinant GluN1/GluN3A receptors, we foresaw that CGP-78608 would represent a powerful tool for revealing endogenous currents mediated by glycine excitatory GluN1/GluN3A receptors in native tissues. The GluN3A subunit is expressed at high levels and almost ubiquitously at early

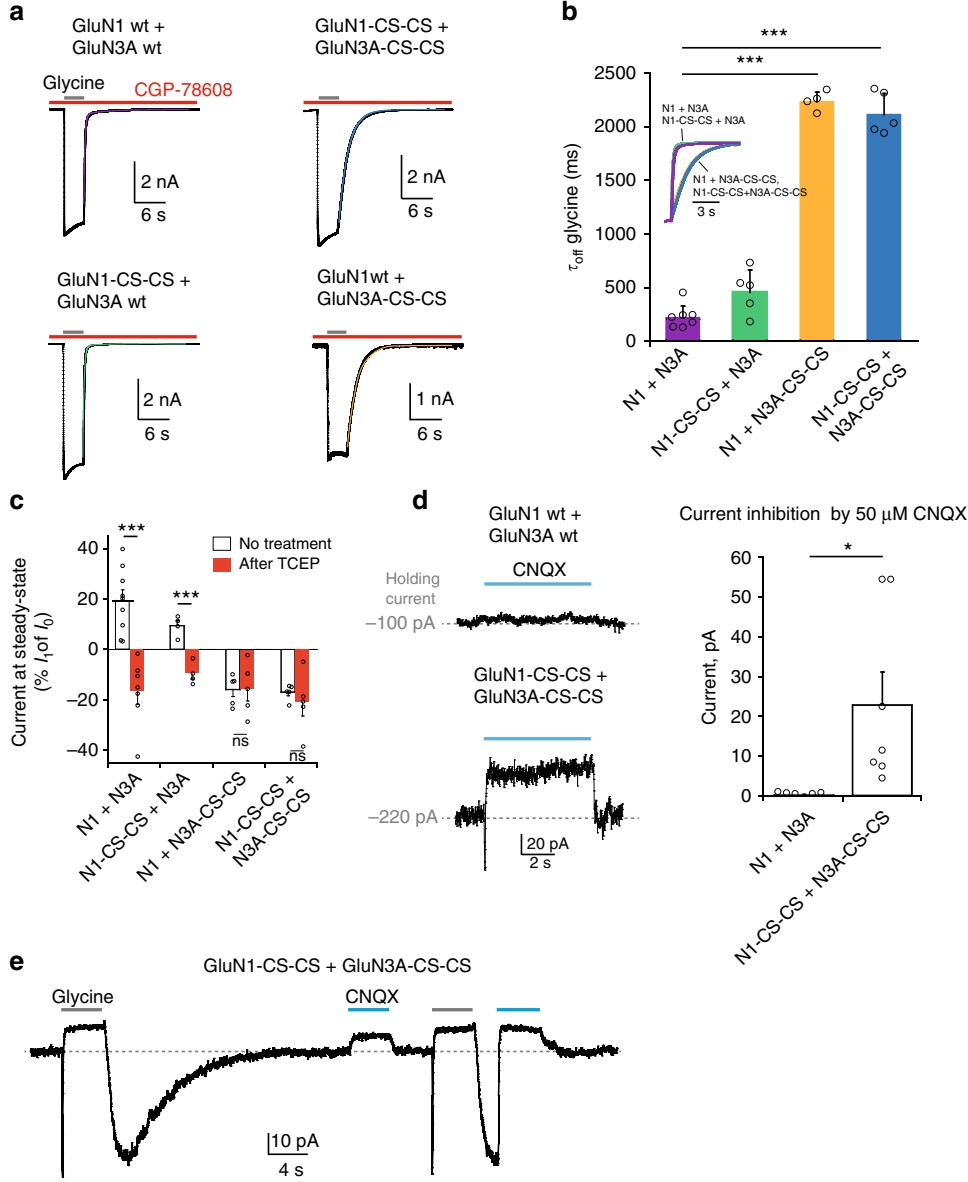

**Fig. 4** High-glycine sensitivity and tonic activation of reduced GluN1/GluN3A receptors. **a** Glycine deactivation kinetics of wild-type and mutant receptors in presence of CGP-78608 (500 nM). Glycine was applied at 100 μM. **b** Quantification of glycine deactivation kinetics. *$P = 0.027$, Student's $t$-test; ***$P < 0.001$, Student's $t$-test ($n = 4$–7). Inset: representative current traces and overlaid fits following glycine washout. **c** Quantification of glycine-induced steady-state outward current shifts following TCEP treatment (see Text). Currents at steady-state ($I_1$) are expressed as percentage of $I_0$ current (see Fig. 2c). ***$P < 0.001$, Student's $t$-test; n.s. $P = 0.95$ and 0.49, respectively, Student's $t$-test; ($n = 4$–9). **d** Effect of CNQX (50 μM) on the holding current (current measured in the absence of glycine) for both wild-type and mutant GluN1-CS-CS/GluN3A-CS-CS receptors. **e** Effect of CNQX (50 μM) on basal and glycine-induced currents carried by GluN1-CS-CS/GluN3A-CS-CS mutant receptors. Note the strong inhibition by CNQX of the tail current (rebound current following glycine removal). Glycine was applied at 100 μM. All recordings were performed in HEK293 cells. Data are mean ± SEM

developmental stages[16,18,20]. We thus explored the ex vivo presence of functional GluN1/GluN3A heterodimers in the hippocampal CA1 region of juvenile mouse slices (P8-P12). We examined the effects of bath applications of CGP-78608 on currents activated by glycine (10 mM) puffed onto voltage-clamped CA1 pyramidal cells (Fig. 5a). To avoid potential activation of inhibitory glycine receptors, these experiments were performed in a cocktail of neurotransmitter receptor inhibitors including the inhibitory glycine receptor inhibitor strychnine (see Methods). In control conditions, glycine induced very small inward currents in both wild-type and GluN3A-knock-out[13] (GluN3A-KO) mice (mean amplitude: 9.3 ± 1.4 pA [$n = 10$] vs 6.0 ± 0.8 pA [$n = 11$], respectively; Fig. 5b, c). Not significantly different ($P = 0.3$;

Mann–Whitney test), these currents are likely due, at least in part, to mechanical artefacts triggered by puffing near the recording electrode.

Bath application of CGP-78608 (1 μM) produced dramatically different effects in WT and GluN3A-KO mice. In wild-type mice, CGP-78608 massively amplified the glycine-induced inward currents to 190.1 ± 58.3 pA ($n = 10$; range: 16.8–640.6 pA; $P = 0.007$ vs control puffs, Wilcoxon signed-rank test), whereas it had no significant effect on currents triggered in GluN3A-lacking animals (mean amplitude of 5.0 ± 0.6 pA [$n = 17$]; $P = 0.9$ vs control puffs in GluN3A-KO animals; $P = 0.00001$ with respect to currents obtained in the presence of CGP-78608 in WT mice; Mann–Whitney test; Fig. 5b, c). Finally, in wild-type mice bath

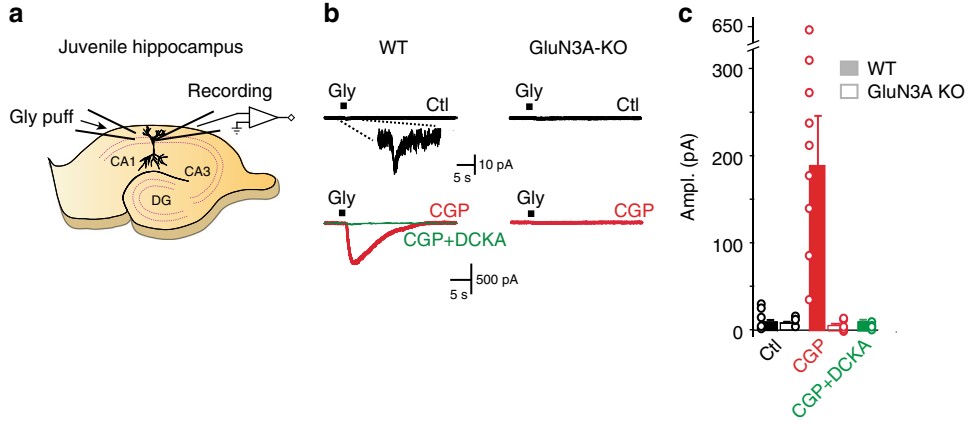

**Fig. 5** GluN1/GluN3A receptors are expressed and functional in juvenile hippocampal slices. **a** Schematic representation of the experimental protocol. Glycine (10 mM) is puffed onto voltage-clamped CA1 cells in acute hippocampal slices from young (P8-12) mice. **b** In control conditions, glycine puffs trigger very small inward currents in both wild-type (WT) and GluN3A-KO mice (upper black traces on both left and right). A typical response obtained in WT mice is shown at larger magnification and time scale in the top left inset. Bath application of CGP-78608 (CGP) leads to massive potentiation of glycine-elicited currents in WT mice, but has no effect in GluN3A-KO animals (bottom red traces). In WT mice, addition of DCKA (500 μM), a GluN1 and GluN3A glycine-binding site antagonist, eliminates currents obtained in the presence of CGP-78608 (bottom left green trace), thus further confirming that GluN1/GluN3A receptors mediate the responses to glycine. **c** Quantification of the experimental results obtained in panel **b**. WT mice, full bars; GluN3A-KO mice, empty bars. Data are illustrated as mean ± SEM

application of DCKA (500 μM), a dual GluN1 and GluN3 glycine-binding site antagonist[10,24], almost fully abolished the currents obtained in the presence of CGP-78608 ($10.2 \pm 2.7$ pA [$n = 9$]; $P = 0.001$; Mann–Whitney test). Altogether, these results provide the first demonstration that glycine excitatory GluN1/GluN3A receptors are expressed and functional in juvenile neurons. Moreover, they highlight the power of CGP-78608 as a new tool compound for detecting GluN1/GluN3A receptor conductances in native systems.

## Discussion

In many aspects, the functional properties and physiological relevance of excitatory glycine GluN1/GluN3 NMDARs remain a conundrum. Here, we provide novel information on the activation, modulation, and pharmacology of GluN1/GluN3A receptors, as well as on their expression in native neurons. We identify a small molecule compound that massively potentiates and extensively transforms GluN1/GluN3A receptor responses. Moreover, we show that GluN1/GluN3A receptors can switch between two distinct modes of activation with very different agonist responsiveness and kinetics, in a redox-dependent manner. We identify an endogenous GluN3A disulfide bridge, as the main molecular actor responsible for this functional plasticity. Finally, we demonstrate that glycine excitatory GluN1/GluN3A receptors can indeed be expressed and be functional in native systems. Hence, these receptors are not just artefacts of heterologous expression systems. Our work thus transforms our current understanding of these underappreciated receptors, and clarifies the long-standing doubts concerning their presence in brain neurons. By providing new ways to manipulate and interrogate GluN1/GluN3A receptors, our work should also facilitate the search for their physiological function.

Building on the differential contribution of the GluN1 and GluN3A subunits to GluN1/GluN3A receptor activation, we discovered that the GluN1 antagonist CGP-78608 acts as a highly potent and powerful potentiator of GluN1/GluN3A receptors. The extent of current potentiation produced by this compound - several orders of magnitude - greatly exceeds effects previously reported by other GluN1 glycine-binding site antagonists. Moreover, CGP-78608 profoundly transformed the time course

of GluN1/GluN3A responses, rendering them much less transient. The enhanced peak amplitude together with the enhanced response stability results in a striking awakening phenotype consisting in a massive overall increase in charge transfer, uncommon in receptor pharmacology. Indeed, many positive allosteric modulators enhance current amplitude without major impact on current time course[47]. Alternatively, drugs can drastically change current shape while having virtually no effect on peak current amplitude, as observed with cyclothiazide on AMPA receptors[48]. The uniqueness of CGP-78608 likely stems from the atypical mechanism of GluN1/GluN3A receptor activation where the same molecule, glycine, serves both as an agonist and a functional antagonist[23,24]. With its high selectivity for GluN1 over GluN3A ABD[11], we suggest that CGP-78608 prevents glycine antagonist effects at GluN1 ABD sites while interfering minimally with the glycine agonist effects at GluN3 ABD sites. In an attempt to formalize this, and other modulatory and gating reactions observed at GluN1/GluN3A receptors, a simple schematic model is proposed (Fig. 6). In this scheme, GluN1/GluN3A receptors can adopt four discrete states along the activation pathway: one resting and one active state, and two desensitized states (Fig. 6a). Upon glycine application, two events resulting in two separate effects occur: receptor activation (i.e. pore opening) by glycine binding to GluN3 sites; receptor inhibition (i.e. entry into desensitized states) following glycine binding to GluN1 sites. These effects can occur either concurrently or sequentially, because GluN3A sites have higher glycine affinity than GluN1 sites[11]. Therefore, exposure to high glycine concentrations leads to small, transient, and rapidly desensitizing responses (Resting- > Active- > Des2; Fig. 6a). Pre-incubation of CGP-78608 prior to glycine application (Pre; Fig. 6b) locks the GluN1 ABD in the open clamshell conformation and prevents glycine binding to GluN1 (Fig. 6b). Little or no CGP-78608 binds to GluN3 given its strong preference for GluN1 (~1000-fold selectivity factor[11]). Following application of saturating glycine, all GluN3 sites are occupied and the receptors fully activate. Meanwhile, little displacement of CGP-78608 from GluN1 sites by glycine occurs, CGP-78608 having much higher affinity as manifested by its long residency time ($\tau_{off}$ in the second timescale, see Fig. 2). Hence, the receptors are stabilized in the active state and avoid desensitization, explaining the massive potentiation produced by CGP-

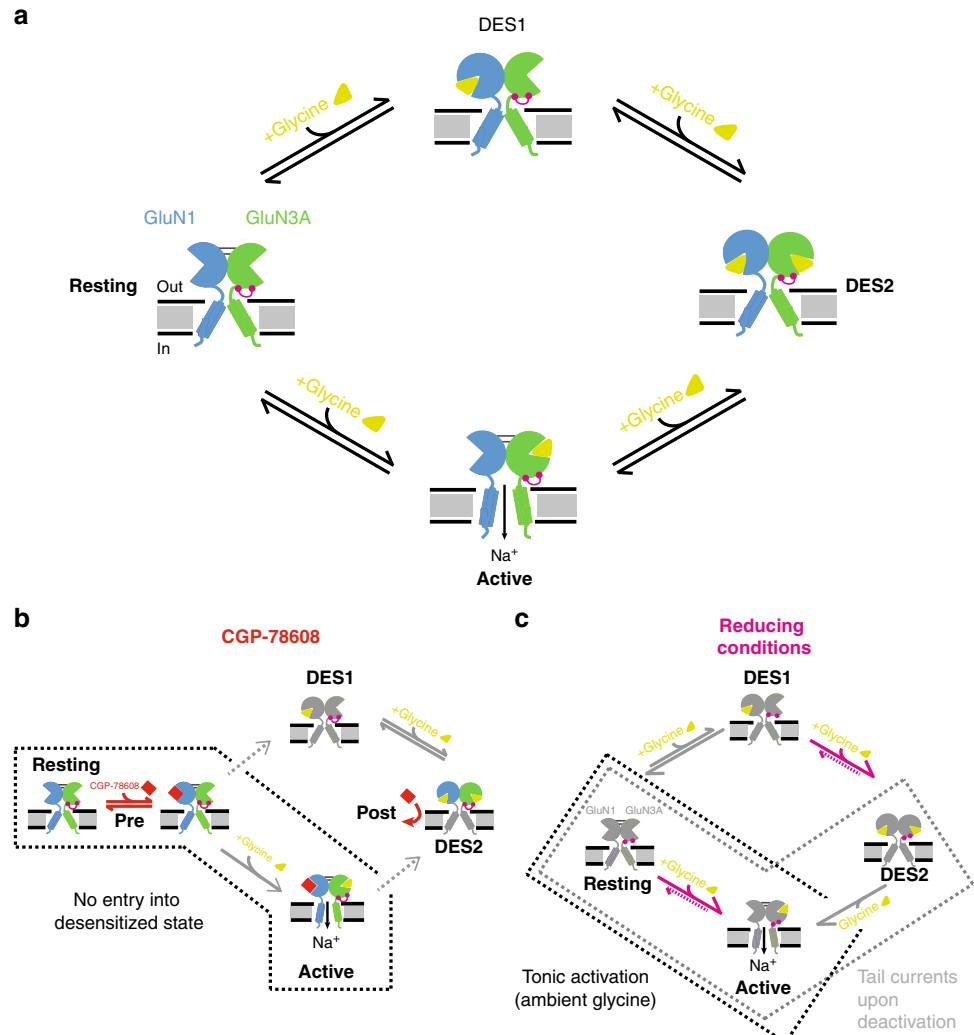

**Fig. 6** Schematic model of GluN1/GluN3A activation and modulation. For clarity, a single GluN1/GluN3A dimer is shown and the NTDs omitted. The upper scheme (**a**) shows receptor activation in control conditions. The GluN3A disulfide bridge conferring high redox sensitivity is highlighted in pink. The lower left scheme (**b**) illustrates activation sequence in presence of CGP-78608. Pre indicates application of CGP-78608 before glycine application while Post indicates the opposite. Pre-incubation with CGP-78608 prior to glycine application enhances receptor activity by preventing glycine binding to GluN1 and subsequent entry into desensitized states. The lower right scheme (**c**) illustrates activation sequence of reduced receptors (GluN3A ABD lower lobe disulfide bridge broken). The main modification is a large increase in the affinity of the GluN3A ABD for glycine resulting in tonic receptor activation by ambient glycine

78608. A different scenario occurs when CGP-78608 is applied after glycine (Post; Fig. 6b). Glycine application shifts most of the receptors into a long-lived glycine-bound desensitized state that would adopt a conformation strongly diminishing CGP-78608 binding. Accordingly, little CGP-78608 potentiation is observed. Increased glycine affinity of GluN1 sites of desensitized receptors likely contributes to this strong state-dependence. Several GluN1/GluN2 modulators also show strong state-dependence (e.g. refs. [49,50]), highlighting the importance of receptor conformational landscape in NMDAR pharmacology. The much greater effects of CGP-78608 compared to other GluN1 antagonists (Fig. 1c, d) likely finds its origin from a unique combination of high potency/high selectivity of CGP-78608 for GluN1 ABD sites. The molecular basis of this unique pharmacological profile still remains to be understood. Binding studies on isolated domains indicate that L689650 discriminates between GluN1 and GluN3 ABDs even better than CGP-78608[11], yet functional effects of L689650 on full-length receptors are much smaller than those produced by CGP-78608 (Fig. 1c, d). The phosphonate group of

CGP-78608, absent in other tested compounds, could make the difference with its specific chemistry (by stabilizing specific ABD/receptor conformations). Ultimately, structures of GluN3A-antagonist complexes should allow determining how CGP-78608 singles out from other GluN1 antagonists.

The robustness of GluN1/GluN3A receptor expression in heterologous systems has been questioned ever since the cloning of GluN3. Several attempts failed to detect noticeable current responses following co-expression of GluN1 and GluN3A subunits, be it in *Xenopus* oocytes[7] or HEK cells[29,51]. In studies examining functional GluN1/GluN3 receptors[23,24,31,52], current responses in control conditions are most often small and transient, complicating their analysis. Our results show that glycine-triggered currents barely detectable in control conditions turn into large and sustained responses in the presence of CGP-78608. This demonstrates that excitatory glycine GluN1/GluN3A receptors express very robustly in heterologous systems, at levels comparable to those of GluN1/GluN2 NMDARs. Our redox experiments further establish that GluN1/GluN3A receptor

activability strongly depends on the receptor's microenvironment. We show that depending on the redox state, the receptor can adopt two radically different gating modes, resulting in different current size and shape. The variability in redox state between experimental settings could thus explain, at least in part, the inconsistency and heterogeneity of GluN1/GluN3A receptor responses previously reported in the literature.

Our redox and mutagenesis experiments reveal that a single endogenous disulfide bridge on the GluN3A subunit – GluN3A-C859-C913 - has major influence on the activation properties of GluN1/GluN3A receptors. This bridge acts as a redox-sensitive gating switch that controls the receptor's glycine affinity and gating kinetics. Under oxidizing conditions, when the two cysteines are cross-linked, the receptor displays a glycine sensitivity in the micromolar range, and a typical activation-desensitization-deactivation pattern upon application of high glycine concentrations. Under reducing conditions, when the disulfide bridge is broken, the gating pattern changes completely. The main modification is a large increase in the GluN3A ABD glycine affinity, estimated between one and two orders of magnitude, and manifested by extremely slow glycine dissociation kinetics from activating GluN3A sites. The enhancement in glycine affinity is such that low ambient glycine concentrations (in the nanomolar range) known to contaminate recording solutions[44–46] are sufficient to occupy high-affinity GluN3A sites (but not low-affinity GluN1 sites) and constitutively activate (a fraction of) GluN1/GluN3A receptors. When high glycine concentrations are applied, all receptors activate and then rapidly desensitize, accumulating in the Des2 state (Fig. 6c). This also concerns receptors activated prior to glycine application, which desensitize following glycine binding to unoccupied GluN1 sites, thus explaining the outward current shift observed in the steady-state. Upon its removal, glycine unbinds rapidly from (low-affinity) GluN1 sites but not from (high-affinity) GluN3A sites. Therefore, as they return to the resting state, receptors transit through the GluN3A-only bound state, the sole active state, accounting for the large tail currents. In such a scenario, tail current deactivation kinetics should closely match glycine dissociation kinetics. This is what is observed (Supplementary Figure 3). Unusual current phenotypes strikingly similar to those described here with constitutive activation, outward shifted steady-state currents and large tail currents have been recently described for kainate receptors harboring clinically-relevant mutations in their transmembrane region[53]. These effects are best explained by profoundly altered gating properties - exquisite sensitivity to ambient agonist (i.e. glutamate) and reopening of receptors as they recover from desensitization[53,54]—as we propose here for glycine excitatory GluN1/GluN3A receptors under reducing conditions. The GluN3A-C859-C913 disulfide bridge occupies a strategic location in GluN3A[12,55], connecting the ABD lower lobe to the TMD ion channel[56,57]. We speculate that its removal releases constraints on the GluN3A ABD clamshell, favoring domain closure, agonist (glycine) binding, and opening of the channel gate. A similar mechanism may also be at play at GluN1 subunits[58], although effects on receptor gating appear much more limited.

By disclosing novel gating and modulation properties of GluN1/GluN3A receptors, our work brings out several important points. First, these receptors are highly plastic in their functionality, capable of operating under various regimes (low/high agonist affinity; transient/persistent activity) according to their environment (redox state). Second, from a molecular and cellular perspective, GluN1/GluN3A receptors are not weakly expressing receptors; on the contrary, they express at high levels, although they appear to be strongly inhibited (or apt to be inhibited) in control conditions. This latter finding is particularly important on

a physiological point of view. The function and identification of native excitatory glycine GluN1/GluN3A receptors have been pending for years. We now reveal that this is not due to a lack of expression but to problems in their detection. Using the GluN1-antagonist CGP-78608 as a novel tool compound to prevent receptor auto-inhibition by glycine, we now show that GluN1/GluN3A receptors are robustly expressed, and electrically functional, in neurons of the juvenile brain. Without the pharmacological assistance of CGP-78608, the GluN1/GluN3A current responses remain virtually undetectable, thus explaining the lack of information on their functional expression until now. Under normal conditions, the activity of excitatory glycine GluN1/GluN3A receptors may be particularly prone to desensitization, or operate under a low tonic regime, as described for persistent inhibitory GABA-A conductances[59]. Our discovery of the powerful awakening effect of CGP-78608 and of the redox switch of GluN1/GluN3A receptors opens new perspectives on the exploration of excitatory glycine receptors and their role in brain development and function.

## Methods

**Ethical statement**. All procedures involving experimental animals were performed in accordance with the European directive 2010/63/EU on the Protection of Animals used for Scientific Purposes, the guidelines of the French Agriculture and Forestry Ministry for handling animals, and local ethics committee guidelines.

**Molecular biology**. The pcDNA3-based plasmid encoding the rat GluN1-4a subunit that was used throughout this work was a kind gift from John Woodward (Medical University of South Carolina, USA). The variant GluN1-4a (named GluN1 herein) was preferred to other GluN1 splice variants because of better assembly into functional diheteromeric GluN1/GluN3 receptors[30]. The pcI_NEO plasmid encoding the rat GluN3A subunit was a kind gift from Isabel Perez-Otaño (Universidad de Navarra, Spain). Site-directed mutagenesis and sequencing procedures were performed as previously described[60].

**Two-electrode voltage-clamp recordings**. Oocytes from *Xenopus lævis* (Xenopus Express, Rennes, France) were used for heterologous expression of recombinant GluN1/GluN3A receptors studied using two-electrode voltage-clamp (TEVC). TEVC recordings were performed 3–4 days following injection. Harvesting and preparation of oocytes was performed as previously described[60] and in the framework of project authorization #05137.02 as delivered by the French Ministry of Education and Research. Each oocyte was co-injected with a mixture of GluN1 and GluN3A cDNA at a concentration of 50 ng/µl and at a 1:1 ratio. The standard external solution contained (in mM): 100 NaCl, 2.5 KCl, 0.3 BaCl$_2$, 5 HEPES, 0.01 DTPA (diethylenetriamine-pentaacetic acid), pH 7.3. Treatments with the reducing agent DTE (dithioerythritol, 5 mM) were performed at room temperature during 15–20 min in a Barth solution. Recordings were performed at −60 mV and at room temperature. Currents were sampled at 100 Hz and low-pass filtered at 20 Hz using an Oocyte Clamp OC-725 amplifier and pClamp 10.5 (Molecular devices). Data analysis was performed using Clampfit 10.5 and Kaleidagraph4 (Synergy Software).

**Whole-cell patch-clamp recordings in HEK cells**. HEK293 cells were used for heterologous expression of recombinant GluN1/GluN3 receptors studied using whole-cell patch-clamp. Cells were obtained from ECACC (European Collection of Authenticated Cell Culture, catalog number: 96121229) and cultured under standard cell culture conditions (95/5% O$_2$/CO$_2$ mixture, 37 °C). Cells were transfected using polyethylenimine with GluN1, GluN3A and GFP plasmids at a 1:3:1 ratio (0.3, 0.9 and 0.3 ng/µl respectively). Recordings were performed 24–48 h following cell transfection. The extracellular solution contained (in mM): 140 NaCl, 2.8 KCl, 1 CaCl$_2$, 10 HEPES and 20 Sucrose (290–300 mOsm), pH adjusted to 7.3 using NaOH. The pipette solution contained (in mM): 115 CsF, 10 CsCl, 10 HEPES and 10 BAPTA (280–290 mOsm), pH adjusted to 7.2 using CsOH. Currents were sampled at 10 kHz and low-pass filtered at 2 kHz using an Axopatch 200B amplifier and Clampex 10.6. Drugs and agonists were applied using a multi-barrel solution exchanger (RSC 200; Bio-logic). Recordings were performed at −60 mV and at room temperature. In experiments with reducing agents, cells were pre-incubated in external solution with Tris(2-carboxyethyl)phosphine (TCEP, 5 mM) or reduced glutathione (GSH, 50 mM) during 20 min.

**Whole-cell patch-clamp recordings in CA1 pyramidal cells**. Ex-vivo electro-physiological experiments were performed on coronal slices from the brains of juvenile wild-type (WT) and GluN3A-KO mice originating from the same litters (8–12 days old). Slices were prepared as described previously[61]. Briefly, mice were anesthetized with isoflurane before decapitation. After isolation, the portion of the

brain containing the hippocampus was placed in bicarbonate-buffered saline (BBS) at 2–5 °C for a few minutes. Slices (300 μm) were then cut using a 7000 smz-2 vibratome (Campden). The slicing procedure was performed in an ice-cold solution containing (in mM): 130 $K^+$- gluconate, 15 KCl, 0.05 EGTA, 20 Hepes, 25 glucose, 1 $CaCl_2$, and 6 $MgCl_2$ supplemented with DL-APV (50 μM). Slices were then transferred for a few minutes to a solution containing (in mM): 225 D-mannitol, 2.5 KCl, 1.25 $NaH_2PO_4$, 25 $NaHCO_3$, 25 glucose, 1 $CaCl_2$, and 6 $MgCl_2$, and finally stored for the rest of the experimental day at 32–34 °C in oxygenated BBS, containing (in mM): 115 NaCl, 2.5 KCl, 1.6 $CaCl_2$, 1.5 $MgCl_2$, 1.25 $NaH_2PO_4$, 26 $NaHCO_3$, and 30 glucose (pH 7.4 after equilibration with 95% $O_2$ and 5% $CO_2$). For all recordings, slices were continuously superfused at 32–34 °C with oxygenated BBS supplemented with NBQX (10 μM; Tocris), SR95-531 (5 μM; Tocris), strychnine (50 μM; Sigma), and TTX (500 nM; Latoxan).

CA1 neurons were recorded from both the ventral and the dorsal hippocampus. Cells were visualized with a combination of Dodt contrast, and an on-line video contrast enhancement. CA1 cells could be easily identified in the red light with which slices were visualized using a CoolSnap HQ2 CCD camera (Photometrics) run by MetaMorph (Universal Imaging). Whole-cell recordings were performed with an EPC-10 double amplifier (Heka Elektronik) run by PatchMaster (Heka). Patch pipettes (resistance 2–3 MΩ) were filled with an intracellular solution containing (in mM): 120 $CsMeSO_3$, 4.6 $MgCl_2$, 10 HEPES, 10 $K_2$-creatine phosphate, 15 BAPTA, 4 $Na_2$-ATP, 0.4 $Na_2$-GTP, 1 QX-314, pH 7.35 with CsOH (~300 mOsm). Series resistance was partially compensated (max 65%), whereas liquid junction potentials were not corrected. Recordings were performed at a holding potential of −60 mV. Puffing pipettes filled with BBS supplemented with glycine (10 mM), NBQX (2 μM), SR95-531 (2 μM), strychnine (50 μM), and D-APV (100 μM), were placed above the slice surface, close to the recording pipettes. Brief (0.5–1 s) puffs were then delivered every 60–120 s via a Picospritzer II (General Valve Corporation).

**Pharmacology and data analysis**. CGP-78608 dose-response curve (DRC) experiments were performed using 100 μM glycine. In these, and most other, experiments, CGP-78608 was systematically applied before agonist perfusion. CGP-78608 DRC was fitted using the following equation: $I_{rel} = 1 + ((a-1)/(1 + ([C]/EC_{50})^{nH})$ were $[C]$ is the compound concentration, $a$ is the maximal potentiation and $n_H$ the Hill coefficient. $EC_{50}$, $a$ and $n_H$ were set as free parameters. Agonist DRCs were fitted with the following Hill equation: $I_{rel} = 1/(1 + (EC_{50}/[A])^{nH})$, were $[A]$ is the agonist concentration, and $EC_{50}$ and $n_H$ set as free parameters. Desensitization and glycine or CGP-78608 washout kinetics were fitted using single exponentials with Clampfit 10.5 analysis software. When comparing the effects of various GluN1 competitive antagonists on GluN1/GluN3A current responses, glycine was used at 1 mM. To insure sufficient GluN1 binding site occupancy, the concentrations of CGP-78608 (500 nM), MDL-29951 (200 nM), 7-CKA (10 μM), and L689560 (1 μM) were set above their Ki values for GluN1 ABD (6.4 nM, 140 nM, 2.6 μM, 29.3 nM; radiolabeled glycine displacement assays on isolated domains[11] or native receptors for MDL-29951[62]). For MDL-29951, in order to insure that larger effects may have not been missed, a higher concentration was also tested (2 μM), yet potentiating factors were found to be smaller than those observed at 200 nM. This is likely due to MDL-29951 binding at GluN3A ABD and inhibiting GluN1/GluN3A receptors.

The compound [(1 S)-1-[[(7-Bromo-1,2,3,4-tetrahydro-2,3-dioxo-5-quinoxalinyl)methyl]amino]ethyl]phosphonic acid hydrochloride (CGP-78608, Tocris) was prepared as a stock solution of 25 mM in 2.2 eq NaOH solution. 2-carboxy-4,6-dichloro-1H-indole-3-propanoic acid (MDL-29951, Cayman Chemical), 7-Chlorokynureic acid (7-CKA, Tocris), trans-2-Carboxy-5,7-dichloro-4-phenylaminocarbonylamino-1,2,3,4-tetrahydroquinoline (L689560, Tocris) and 6-Cyano-7-nitroquinoxaline-2,3-dione (CNQX, Alomone) were prepared in DMSO at stock concentrations of 2 mM, 10 mM, 2 mM, and 50 mM, respectively. 5-7-Dichlorokynurenic acid (DCKA, Tocris) stocks (50 mM) were prepared in a 1 eq NaOH solution. Glycine (1 mM; Sigma), strychnine (10 mM; Sigma), 6-Imino-3-(4-methoxyphenyl)-1(6 H)-pyridazinebutanoic acid hydrobromide (SR95-531; 10 mM; Tocris), 2,3-Dioxo-6-nitro-1,2,3,4-tetrahydrobenzo[f]quinoxaline-7-sulfonamide disodium salt (NBQX; 10 mM; Tocris), D-(-)-2-Amino-5-phosphonopentanoic acid (D-APV; 50 mM; Tocris) stocks were all prepared in water. Finally, Tetrodotoxin (TTX; 500 μM; Latoxan) was aliquoted in an acetic acid (2%) water solution.

**Statistical analysis**. Data are presented as mean ± standard error of the mean (SEM). Sample number ($n$) refers either to the number of recorded oocytes or HEK293 cells (Figs. 1–4), or to the number of recorded CA1 cells (Fig. 5). Unless otherwise stated, two-sided statistical analysis of the data shown in Figs. 1–4 was conducted using either two sample Student's $t$-test or the Mann–Whitney test when Equal Variance test failed. Data obtained from different conditions in Fig. 5 were compared statistically either with Mann–Whitney or Wilcoxon signed-rank test, as indicated in the text. Tests were performed using either SigmaPlot (Systat Software) or house-built routines in Igor (Wavemetrics). Statistical significance is reported in the figures according to the following symbols *, ** and ***, which indicate $P$ values below 0.05, 0.01, and 0.001, respectively. n.s. = not significant.

**Data availability**

The datasets generated and analysed during the current study are available from the corresponding authors on reasonable request.

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

## Acknowledgements
This work was supported by the French government (Investissements d'Avenir ANR-10-LABX-54 MEMO LIFE and ANR-11-IDEX-0001-02 PSL* Research University), Agence Nationale de la Recherche (ANR grant GluBrain3A to M.A.D. and P.P.), and the European Research Council (ERC Advanced Grant #693021 to P.P.). We thank Nobuki Nakanishi and Stuart Lipton (Scintillon Institute, San Diego, CA, USA) for providing the GluN3A-KO mouse line. We also thank Boris Barbour and Mariano Casado for critical reading of the manuscript.

## Author contributions
T.G., M.A.D., and P.P. designed the project. T.G. performed all the experiments on recombinant receptors and S.A.G. performed the experiments on hippocampal slices. M.D. provided technical support in molecular biology and cell culture. T.G., S.A.G., and P.P. analyzed the data, and T.G., M.A.D., and P.P. wrote the manuscript. M.A.D. and P.P. supervised the project.

## Additional information

**Competing interests:** The authors declare no competing interests.

