## [Peer Review File · Nature Communications]

Reviewers' comments:

Reviewer #1 (Remarks to the Author):

This clever paper was a joy to read and presents very interesting results. The N3 subunits of NMDA receptors have languished under the perception of being hard to work with, hard to express and hard to understand since their cloning. They have unique properties but have proven hard to measure in electrophysiological assays mainly because their current responses are so small and unstable.

Typically careful work from the IBENS group work reveals two ways to massively increase a) current responses or b) sensitivity to agonist. The results will have a big impact in facilitating the study of these subunits. The experiments also provide a lot of mechanistic information, and introduce the unprecedented response of "constitutive activity" (due to background glycine) upon reduction. On a more general level, "the potentiation by an antagonist" model that the authors introduce is quite novel, and might perhaps be discussed by the authors more widely in the context of receptor pharmacology.

The paper is very nicely written and I have only minor points of interpretation, grammar and typos.

Minor comments

line 54 "Incomparably less is known regarding NMDARs" - sounds odd, the comparison is obvious. Perhaps, much?

line 61 "before progressively declining" : in abundance?

line 71 "First, they would constitute a new type of glycine excitatory receptor" Well, they do, if they are actually expressed anywhere is the question. Also, the authors sometimes refer to "excitatory glycine N1 N3 NMDARs" (e.g. lines 170, 344) - so the case appears to be closed, and "would" is not the right sense. Perhaps this idea is more precisely phrased this as a question? "Do n1N3 actually function as excitatory glycine receptors in neurons?"

line 92 "awakens" or "wakes up" (same on line 645)

line 147 "competitive interplay" - I feel it is unwise to use competition in this sense. You are talking about the relation between two separate binding sites - the mechanism is not clear and it's almost certainly not a kind of exclusive, competitive binding, whereby antagonist can bind to N1 or glycine can bind to N3 but not both.

line 165 - "another expression system widely used to study membrane receptors" : it's more relevant that a lot of work on N1N3 was done in oocytes, and that some deviations between HEK and oocyte results were previously observed for GluRs. I would be more specific here.

line 250 "GluN3A"

line 278 "However, it echoes surprisingly well the 40.4 nM glycine affinity found by Yao and Mayer using radiolabeled ligand assay on the isolated GluN3A ABD." - A nice point, but I think that this is a coincidence. I do not believe that the relevant disulphide bond was reduced in these experiments. Isolated LBD binding assays always tend to give very tight binding.

line 282: "glycine excitatory GluN1/GluN3 NMDARs" wrong order of adjectives.

Figure 3C You might mention that this kind of rebound current is thought to happen for AMPA-receptors with TARPs:

Lu, H. W., Balmer, T. S., Romero, G. E., and Trussell, L. O. (2017). Slow AMPAR Synaptic Transmission Is Determined by Stargazin and Glutamate Transporters. *Neuron* 96, 73-80.e4.

Seems like an interesting connection - also recovery from desensitization through the open state.

Andrew Plested

Reviewer #2 (Remarks to the Author):

This paper reports a mechanistic analysis of the potentiating action of CGP-78608 on recombinant GluN1/GluN3A 'excitatory' glycine receptors. This drug is then used as a tool to investigate how the oxidation state of a structurally conserved disulfide bond in the ligand binding domain of GluN1 and GluN3 subunits regulates receptor activation and desensitization.

Because a substantial body of prior work by other groups has already established that the GluN1 and GluN3 subunits have different roles in the activation and desensitization of GluN1/GluN3A heteromeric receptors, essentially identical to those reported here, including activation of ion channel gating by binding of glycine to GluN3A subunit alone, biphasic agonist dose response curves, the description of tail currents commonly observed with slow perfusion systems e.g. for *Xenopus* oocytes following removal of agonists, and modification of gating by the GluN1 F484A mutant, the novelty of the current paper lies in (i) the unusually large effect of CGP-78608, with the proposition that it could be used as a tool to investigate native GluN1/GluN3A receptors and (ii) identification of the GluN3A LBD disulfide bond as a powerful regulator of receptor activity. Although gating models for GluN1/GluN3A gating are presented in Figure 5, these are largely descriptive, with no attempts to perform quantitative experiments to measure glycine concentration response curves, for example for WT receptors in the presence of CGP-78608, and for the GluN3A-CS-CS mutant; these could be easily performed. More challenging, but not impossible, are glycine concentration response curves for peak currents, measured with rapid perfusion systems. Without this data the model shown in Figure 5 lacks rigor, a problem that has confounded study of 'excitatory' glycine receptors following their discovery.

Given this background of substantial prior work, and the modest advance in better describing the gating of GluN1/GluN3A receptors, the study would have immensely greater impact if experiments were performed using CGP-78608 to demonstrate expression of and to begin characterize native GluN3A containing receptor assemblies, as the authors propose in the discussion. This could be done using either brain slices from developmental stages where GluN3A expression is still high, or spinal cord slices to study adult motoneurons which express GluN3A. Without this, in its present form, the study is an interesting, but incremental advance that will be of interest to a largely specialized audience, and perhaps more appropriate for a specialized journal, such as *Molecular Pharmacology*.

Additional points.

1) It is far from clear why CGP-78608 exhibits the unusual effects reported here, compared to other GluN1 and GluN3A antagonists. Specifically, the ratio of K_d s for binding to isolated GluN1 and GluN3A LBDs reported by Yao & Mayer (ref 11) is actually much greater for L689560 than CGP-78608, with a K_d for GluN1 only 5-fold lower, yet the functional profiles of these two drugs differ substantially. Would a 5-fold faster rate constant for dissociation of L689560 from GluN1 be sufficient to account for the different behavior? The lack of available structures for GluN3A LBD antagonist complexes makes understanding this difference a challenge, but it is notable that CGP-78608 contains a phosphonate group; perhaps this, or other structural properties contribute to its unique functional profile.

2) Although both the F484A mutant and CGP-78608 act by preventing binding of glycine to the GluN1 subunit, their functional profiles differ. With CGP-78608 there is modest slow desensitization (Fig 1A) while with F484A there is rapid and stronger desensitization (Fig S1B). Why this is so should be discussed.

3) The analysis of glycine responses following application of reducing agents, and the text on lines 184-185, is obtuse. Before application of glycine there is a steady inward current, due to activation of GluN1/GluN3A by ambient glycine; this current is decreased, due to desensitization, when glycine is applied at higher concentrations. In healthy cells the holding current in absence of GluN1/GluN3A should be close to zero, and it should be possible to quantify the resting current directly and not in terms of the ratios of I_h/I_1 presented in Figure 3C and discussed on lines 256-259.

Minor points.

Line 72: glycine is extremely well established as the primary inhibitory transmitter in the spinal cord (e.g. the convulsant action of strychnine). Rephrase "normally thought of as an inhibitory neurotransmitter" and replace reference 10 by citation of classical literature establishing the role of glycine in inhibitory synaptic transmission.

Line 113: it would be useful to give in addition the range of potentiation since a mean \pm SEM of 335 ± 266 with $n=11$ implies a huge variation; presumably this is due to a spread in the upper and not lower boundary of the extent of potentiation?

Line 161: without measurement of full concentration response curves it is not possible to state that D-ser acts as a partial agonist.

Figure 3E: the majority of experiments were performed using TCEP; why was DTE used for HEK cell recordings?

NCOMMS-18-14769
Revised manuscript
Reply to reviewers

We thank the two reviewers for their careful reading and insightful comments on our manuscript. Following the reviewers' suggestions, we have made substantial modifications to our work. Most particularly, we performed additional experiments on juvenile brain slices showing for the first time that glycine excitatory GluN1/GluN3A receptors are functionally expressed in neurons. This key new result was made possible through the use of the compound CGP-78608 that we describe in the manuscript. We can now firmly state that CGP-78608 is, and will be, an essential tool to reveal and work on native glycine excitatory GluN3A-containing receptors. In addition to these critical new data, the revised manuscript also includes additional experimental results on recombinant GluN1/GluN3A receptors (including glycine concentration-response curves), and several modifications in the main Text to clarify points at various locations. We believe that altogether these revisions lead to a greatly improved manuscript that will be of broad interest to a large audience.

Reviewer #1 (Remarks to the Author):

This clever paper was a joy to read and presents very interesting results. The N3 subunits of NMDA receptors have languished under the perception of being hard to work with, hard to express and hard to understand since their cloning. They have unique properties but have proven hard to measure in electrophysiological assays mainly because their current responses are so small and unstable.

Typically careful work from the IBENS group work reveals two ways to massively increase a) current responses or b) sensitivity to agonist. The results will have a big impact in facilitating the study of these subunits. The experiments also provide a lot of mechanistic information, and introduce the unprecedented response of "constitutive activity" (due to background glycine) upon reduction. On a more general level, "the potentiation by an antagonist" model that the authors introduce is quite novel, and might perhaps be discussed by the authors more widely in the context of receptor pharmacology.

The paper is very nicely written and I have only minor points of interpretation, grammar and typos.

Minor comments

line 54 "Incomparably less is known regarding NMDARs" - sounds odd, the comparison is obvious. Perhaps, much?

We have replaced 'incomparably' by 'much'.

line 61 "before progressively declining" : in abundance?

'in abundance' has been added at the end of the sentence.

line 71 "First, they would constitute a new type of glycine excitatory receptor" Well, they do, if they are actually expressed anywhere is the question. Also, the authors sometimes refer to

"excitatory glycine N1 N3 NMDARs" (e.g. lines 170, 344) - so the case appears to be closed, and "would" is not the right sense. Perhaps this idea is more precisely phrased this as a question? "Do n1N3 actually function as excitatory glycine receptors in neurons?"

The word 'would' was indeed inappropriate and has been deleted. And, we now provide the first evidence that excitatory glycine GluN1/GluN3A NMDA receptors are indeed expressed in native neuronal tissues (see reply to reviewer 2).

line 92 "awakens" or "wakes up" (same on line 645)

Corrected ('awakens'), thank you.

line 147 "competitive interplay" - I feel it is unwise to use competition in this sense. You are talking about the relation between two separate binding sites - the mechanism is not clear and it's almost certainly not a kind of exclusive, competitive binding, whereby antagonist can bind to N1 or glycine can bind to N3 but not both.

It is right that the word 'competitive' is inappropriate since multiple distinct binding sites are in play. What we had in mind is the competitive (i.e. exclusive) interactions between the drug compounds (the antagonists) and the GluN1 glycine site. Yet, the compounds may also bind to the GluN3A glycine site (although with a much weaker affinity), rendering the possible interactions quite complex (some being truly competitive, other not). That is why we mentioned both subunits, GluN1 and GluN3A, in this sentence. To avoid any confusion and misusage of the word 'competitive', we have deleted the word 'competitive' and replaced it by 'complex'.

line 165 - "another expression system widely used to study membrane receptors" : it's more relevant that a lot of work on N1N3 was done in oocytes, and that some deviations between HEK and oocyte results were previously observed for GluRs. I would be more specific here.

We have modified the sentence to make clearer that the effects of CGP-78608 are observed, and equally robust, regardless of the expression system used.

line 250 "GuN3A"

Corrected, thank you for spotting this typo.

line 278 "However, it echoes surprisingly well the 40.4 nM glycine affinity found by Yao and Mayer using radiolabeled ligand assay on the isolated GluN3A ABD." - A nice point, but I think that this is a coincidence. I do not believe that the relevant disulphide bond was reduced in these experiments. Isolated LBD binding assays always tend to give very tight binding.

The disulfide bridge in Yao and Mayer (2006) is indeed present, but the tension exerted by the linker between the ABD and the TMD M4 segment is very likely missing (especially as the last amino acid of the crystal structure is C913, the one involved in the disulphide bond; see Figure 3A). In other words, the isolated ABDs studied by Yao and Mayer likely adopt a 'relaxed' state mimicking the impact of disulfide bond breakage in intact functional receptors.

line 282: "glycine excitatory GluN1/GluN3 NMDARs" wrong order of adjectives.

Corrected.

Figure 3C You might mention that this kind of rebound current is thought to happen for AMPA-receptors with TARPs:

Lu, H. W., Balmer, T. S., Romero, G. E., and Trussell, L. O. (2017). Slow AMPAR Synaptic Transmission Is Determined by Stargazin and Glutamate Transporters. *Neuron* 96, 73-80.e4.

Seems like an interesting connection - also recovery from desensitization through the open state.

This is an interesting point that had escaped our attention. Reading carefully the literature on rebound currents described on other iGluRs, we stumbled on a study (Guzman et al., *Neurology Genetics*, 2017) even more striking than that of Lu et al. describing mutant heteromeric kainate receptors (GluK2(A657T)/GluK5) with a phenotype conspicuously similar to the one that we describe on reduced excitatory glycine GluN1/GluN3A receptors: upon agonist (glutamate) application, currents transiently activate (peak current) before desensitization to a steady-state current that is more positive than the baseline holding current, while upon agonist removal a rebound current is elicited (see Figure 3C of Guzman et al.). Interestingly, the authors interpret these data as indicating that mutant kainate receptors are exquisitely sensitive to glutamate resulting in constitutive activation by trace levels of glutamate. Upon glutamate removal, a rebound current is observed likely because the receptors reopen as they recover from desensitization (as described also in Lu et al.). This is exactly what we propose for reduced GluN1/GluN3A receptors. These receptors would be 'super' sensitive to glycine and constitutively activated by trace levels of glycine known to contaminate recording solutions. Rebound currents upon agonist (glycine) removal would then occur because of (fast) dissociation of glycine from 'inhibitory/desensitizing' GluN1 sites and transient re-opening of GluN3A-only glycine bound GluN1/GluN3A channels. It is satisfactory to note that gain-of-function modifications from different origin (either through mutagenesis in the TMD region in the case of kainate receptors or redox modification in the ABDs in the case of GluN1/GluN3 receptors) can result in shared mechanisms and common phenotypes across various receptor types. In the revised manuscript, we now extend the discussion to other receptor types and highlight how the work by Guzman and colleagues as well as Lu and colleagues provide further support to our interpretation and conclusions (Discussion section, p14).

Reviewer #2 (Remarks to the Author):

This paper reports a mechanistic analysis of the potentiating action of CGP-78608 on recombinant GluN1/GluN3A 'excitatory' glycine receptors. This drug is then used as a tool to investigate how the oxidation state of a structurally conserved disulfide bond in the ligand binding domain of GluN1 and GluN3 subunits regulates receptor activation and desensitization.

Because a substantial body of prior work by other groups has already established that the GluN1 and GluN3 subunits have different roles in the activation and desensitization of GluN1/GluN3A heteromeric receptors, essentially identical to those reported here, including activation of ion channel gating by binding of glycine to GluN3A subunit alone, biphasic

agonist dose response curves, the description of tail currents commonly observed with slow perfusion systems e.g. for *Xenopus* oocytes following removal of agonists, and modification of gating by the GluN1 F484A mutant, the novelty of the current paper lies in (i) the unusually large effect of CGP-78608, with the proposition that it could be used as a tool to investigate native GluN1/GluN3A receptors and (ii) identification of the GluN3A LBD disulfide bond as a powerful regulator of receptor activity. Although gating models for GluN1/GluN3A gating are presented in Figure 5, these are largely descriptive, with no attempts to perform quantitative experiments to measure glycine concentration response curves, for example for WT receptors in the presence of CGP-78608, and for the GluN3A-CS-CS mutant; these could be easily performed. More challenging, but not impossible, are glycine concentration response curves for peak currents, measured with rapid perfusion systems. Without this data the model shown in Figure 5 lacks rigor, a problem that has confounded study of 'excitatory' glycine receptors following their discovery.

Given this background of substantial prior work, and the modest advance in better describing the gating of GluN1/GluN3A receptors, the study would have immensely greater impact if experiments were performed using CGP-78608 to demonstrate expression of and to begin characterize native GluN3A containing receptor assemblies, as the authors propose in the discussion. This could be done using either brain slices from developmental stages where GluN3A expression is still high, or spinal cord slices to study adult motoneurons which express GluN3A. Without this, in its present form, the study is an interesting, but incremental advance that will be of interest to a largely specialized audience, and perhaps more appropriate for a specialized journal, such as *Molecular Pharmacology*.

According to these comments, we performed novel experiments on mouse brain slices to explore the functional expression of GluN1/GluN3A receptors in native neuronal tissue. The reviewer is right that demonstrating functional expression of these receptors in native tissue would represent a major step forward. Because GluN3A expression peaks early on during postnatal development and shows a widespread distribution ~10 days after birth, we recorded from acute hippocampal slices from P8-P12 mice. Both WT and GluN3A-KO mice were used. As illustrated in the new Figure 5, we show that in WT animals and in the presence of CGP-78608, but not in its absence, puffing glycine elicits large (>100 pA) inward currents. These currents are inhibited by DCKA, a known antagonist of GluN1/GluN3A receptors. Moreover, we show that these currents are absent in GluN3A-KO even in the presence of CGP-78608. These data are important for several reasons: first, they provide the first convincing demonstration that glycine excitatory GluN1/GluN3A receptors are expressed and functional in native neuronal tissue (juvenile hippocampus); second, they highlight the power of CGP-78608 as a unique tool compound for detecting GluN1/GluN3A receptor activity in the brain; third, they reshape our understanding of glycinergic neurotransmission and open new perspectives on the exploration of excitatory glycine receptors and their role in brain development and function. These new data have been introduced in the revised manuscript as a new main figure (Figure 5) and are discussed in a dedicated paragraph in the Discussion section (p15). The Methods section also contains a new section to describe the slice experiments. Finally, the author list has been modified to include Sarah Abi Gerges, the person who performed these experiments.

Regarding gating of GluN1/GluN3A receptors, we agree with the reviewer that the model presented in the last figure is simplistic and does not provide a detailed quantitative gating scheme. Yet, we think it represents an interesting and informative piece of information for the reader to apprehend how CGP-78608 and redox agents interact with GluN1/GluN3A receptors to perturb their gating mechanisms. Because GluN1/GluN3A receptors display complex and very fast gating kinetics (in particular much faster desensitization kinetics than 'conventional' GluN1/GluN2 NMDARs), proper quantification of gating parameters would require a very fast perfusion system as developed for (fast-desensitizing) AMPA receptors

(theta tube on a piezo system) and allowing solution exchange within less than a millisecond. Such a system is not available in our lab yet but we are willing to implement it in the near future. Together with kinetic modelling, our ambition is to identify a gating scheme that would describe the key characteristics of the macroscopic current responses of GluN1/GluN3A receptors. We believe this represents a full project by itself and goes beyond the current work that describes several novel features of these receptors. We are also planning to perform single-channel recordings to directly evaluate the biophysical (kinetic) properties of GluN1/GluN3A receptor channels.

In order to get better assessment of GluN1/GluN3A activation parameters, and as requested by the reviewer, we performed glycine concentration-response curves on WT receptors with and without CGP-78608 and on mutant receptors lacking the endogenous ABD lower lobe disulfide bridges (GluN1-CS/GluN3A-CS). These experiments were performed on *Xenopus* oocytes for comparison purposes with past reports (see Madry et al., BBRC 2007 and Awobuluyi et al., Mol Pharmacol, 2007), and are presented in the revised Supplementary Figure 2. In control conditions, WT GluN1/GluN3A receptors display glycine sensitivity in the low μM range (glycine $\text{EC}_{50} = 7.1 \mu\text{M}$, peak responses), in good agreement with previous estimates (see Madry et al., 2007). In the presence of CGP-78608 (500 nM), in conditions where glycine-induced responses are several orders of magnitude larger, glycine sensitivity is moderately decreased ($\text{EC}_{50} = 39 \mu\text{M}$). This ~5-fold rightward shift in agonist sensitivity can be explained either by direct competition between glycine and CGP-78608 binding to GluN3A ABD sites (which display low μM sensitivity for CGP-78608, 3-orders of magnitude lower than GluN1 ABD sites; see Yao and Mayer, 2006), or by an indirect (i.e. allosteric) effect of CGP-78608 binding at GluN1 ABD sites onto GluN3A ABD glycine sites. To distinguish between these two possibilities, we repeated the concentration-response curve experiment with a 10-fold lower CGP-78608 concentration (50 nM). As shown in the panel C of the revised Supplementary Figure 2, in this condition, glycine EC_{50} (40 μM) is almost identical to that measured with 500 nM CGP-78608, strongly suggesting that CGP-78608 decreases glycine sensitivity of GluN1/GluN3A receptors through an inter-subunit allosteric effect between GluN1 and GluN3A ABD sites (rather than a direct competitive effect with glycine on GluN3A ABD sites). Finally, we performed glycine-concentration response curves on mutant GluN1-CS/GluN3A-CS receptors recorded in the presence of CGP-78608 (500 nM). Compared to WT receptors recorded in the same conditions, glycine sensitivity was increased by >10-fold ($\text{EC}_{50} = 3 \mu\text{M}$). This result (increase in agonist sensitivity) is fully consistent with our conclusions, based on other experimental evidence (glycine deactivation kinetics, presence of tonic currents; see Figure 4), that reduced GluN1/GluN3A receptors show greatly enhanced agonist sensitivity. These new results are now included in the revised manuscript (see Text p7 & p9 and new Supplementary Figure 2).

Additional points.

1) It is far from clear why CGP-78608 exhibits the unusual effects reported here, compared to other GluN1 and GluN3A antagonists. Specifically, the ratio of K_d s for binding to isolated GluN1 and GluN3A LBDs reported by Yao & Mayer (ref 11) is actually much greater for L689560 than CGP-78608, with a K_d for GluN1 only 5-fold lower, yet the functional profiles of these two drugs differ substantially. Would a 5-fold faster rate constant for dissociation of L689560 from GluN1 be sufficient to account for the different behavior? The lack of available structures for GluN3A LBD antagonist complexes makes understanding this difference a challenge, but it is notable that CGP-78608 contains a phosphonate group; perhaps this, or other structural properties contribute to its unique functional profile.

We thank the reviewer for raising this interesting point. Indeed, according to Yao and Mayer (J Neurosci, 2006), the compound L689560 discriminates between GluN1 and GluN3 LBDs

'better' than CGP-78608 (8-fold difference between the two compounds). Hence, one could have expected L689560 to mediate effects on GluN1/GluN3A receptor activity at least as large as CGP-78608. Our functional data clearly show that this is not the case and that CGP-78608 is very unique among the panel of compounds tested in transforming GluN1/GluN3A responses. Why CGP-78608 is so much more efficient than L689560 is unknown. It is important to note, however, that Yao & Mayer have measured binding affinities on isolated LBDs, while we record from intact (i.e. full length) membrane-imbedded receptors. It is conceivable that on full-length receptors, L689560 has a lower discriminative power than on isolated LBDs thus antagonizing receptor activity through GluN3A LBD binding. As mentioned, we lack 3D structures of GluN3A ABD antagonist complexes to help understand the molecular basis of the unique functional profile of CGP-78608. Yes, the phosphonate group, with its specific size and chemistry and lacking in other compounds (L689560, MDL, 7-CKA) could make the difference (by stabilizing specific ABD/receptor conformations). In the revised manuscript, we now discuss these aspects (Discussion section, p13, end of 1st paragraph).

2) Although both the F484A mutant and CGP-78608 act by preventing binding of glycine to the GluN1 subunit, their functional profiles differ. With CGP-78608 there is modest slow desensitization (Fig 1A) while with F484A there is rapid and stronger desensitization (Fig S1B). Why this is so should be discussed.

The most likely reason is that the mutation GluN1-F484A does not abolish glycine binding as much as CGP-78608 does (at 500 nM). Despite strongly reducing glycine affinity, it has already been shown that the mutation GluN1-F484A does not fully abolish glycine binding to GluN1 (see Kvist et al., *Neuropharmacology*, 2013). Thus, when added at sufficiently high concentrations, glycine can occupy mutant GluN1 sites and inhibit (i.e. desensitize) GluN1/GluN3A receptors (our data, and see also Figure 1 of Kvist et al.). In contrast, with the high-affinity ligand CGP-78608, receptor inhibition is minimal because the vast majority of inhibitory GluN1 ABD sites are 'blocked' by CGP-78608 even at high glycine concentrations. Interestingly, a second mutation can be introduced in GluN1 ABD (GluN1-F484A-T518L) to fully abolish glycine binding (up to few tens of mM concentrations; Kvist et al., 2013). We tested this double mutant in HEK cells but unfortunately its expression rate is very low, rendering its study problematic. Accordingly, we chose to work with the single GluN1-F484A mutant. We would also like to stress that the mutation GluN1-F484A was primarily used to show that the massive potentiating effects of CGP-78608 on GluN1/GluN3A responses required ligand binding to the GluN1 ABD.

In the revised manuscript, we have added a sentence to the legend of Supplementary Figure 1B mentioning that the fast desensitization observed at GluN1-F484/GluN3A receptors likely stems from residual glycine binding to the mutant GluN1 ABD. Moreover, we refer to the Kvist et al. (2013) paper.

3) The analysis of glycine responses following application of reducing agents, and the text on lines 184-185, is obtuse. Before application of glycine there is a steady inward current, due to activation of GluN1/GluN3A by ambient glycine; this current is decreased, due to desensitization, when glycine is applied at higher concentrations. In healthy cells the holding current in absence of GluN1/GluN3A should be close to zero, and it should be possible to quantify the resting current directly and not in terms of the ratios of I_h/I₁ presented in Figure 3C and discussed on lines 256-259.

First, we have modified the text in the Results section (p7) to make clearer the effects of reducing agents on glycine responses. We focused most particularly on the section describing the 'steady-state' current following glycine application. We now state: 'When

glycine was applied, the peak current from reduced GluN1/GluN3A receptors desensitized to a steady-state current that was *more positive* than the baseline holding current.' We also modified lines 256-257 accordingly. Furthermore, at several locations, we replaced 'offset steady-state current', which is not very clear, by 'outward shifted steady-state current', which is more precise and directly pointed to the red bars of Figure 3C.

Second, we performed a novel series of experiments to quantify the resting currents mediated by reduced GluN1/GluN3A receptors tonically activated by ambient glycine. To detect and quantify constitutive activation, we measured the effects of CNQX (50 μ M), an antagonist of GluN1/GluN3A receptors, on the holding current amplitude. For cells expressing wild-type GluN1/GluN3A receptors, CNQX barely changed the holding current (outward current displacement of 0.6 ± 0.06 pA, [n=6]), as expected if the receptors are not tonically active. In contrast, for cells expressing GluN1-CS-CS/GluN3A-CS-CS receptors (receptors lacking the endogenous ABD lower lobe disulfide bond), CNQX strongly affected the holding current, displacing its amplitude by 22.9 ± 8.3 pA (n=7) from baseline, thus indicating a significant level of constitutive activation. These novel data, which provide a more direct measurement of the redox-dependent gating switch, are included in the revised manuscript as a novel panel in Figure 4 (panel D). We also modified the text accordingly (Results section, p9). Note that in the initial submission, we had already performed a CNQX experiment on mutant (i.e. reduced) receptors (showing constitutive receptor activation; old Fig 4D), yet we had no side-by-side comparison with non-reduced wild-type receptors and no quantification.

Minor points.

Line 72: glycine is extremely well established as the primary inhibitory transmitter in the spinal cord (e.g. the convulsant action of strychnine). Rephrase "normally thought of as an inhibitory neurotransmitter" and replace reference 10 by citation of classical literature establishing the role of glycine in inhibitory synaptic transmission.

We have rephrased this sentence (Results section p3) and added three references related to the (well-established) role of glycine in inhibitory synaptic transmission (one original reference, Curtis et al., Nature, 1967; and two comprehensive review articles, Legendre et al., Cell Mol Life Sci, 2001; Bowery & Smart, Brit J Pharmacol, 2006).

Line 113: it would be useful to give in addition the range of potentiation since a mean \pm SEM of 335 ± 266 with n=11 implies a huge variation; presumably this is due to a spread in the upper and not lower boundary of the extent of potentiation?

As required, we added the range of potentiation in the Text (50-296 and 106-876 for peak and steady-state currents, respectively; Results section p5). The large variability observed largely stems from the difficulty in precisely measuring current intensities in control condition (prior CGP-78608 application) given their very small amplitudes. This is most particularly critical for steady-state (desensitized) currents, which can be of minuscule amplitude (and barely measurable) in control conditions. Yet, in every single cell tested, CGP-78608 drastically augmented current amplitudes, both at peak and steady-state.

Line 161: without measurement of full concentration response curves it is not possible to state that D-ser acts as a partial agonist.

The sentence has been modified to (Results section p6): 'Using CGP-78608, we also confirmed that D-serine (up to 500 μ M) triggered currents of smaller amplitude than those elicited by glycine (100 μ M) (Supplementary Figure 1C), in agreement with D-serine having lower efficacy than glycine at GluN1/GluN3A receptors.'

Figure 3E: the majority of experiments were performed using TCEP; why was DTE used for HEK cell recordings?

In the current work, TCEP was used on HEK cells and DTE on *Xenopus* oocytes. We usually employ DTE in most of our redox experiments when working with *Xenopus* oocytes (see, for instance, Gielen et al., *Neuron*, 2008). When patching HEK cells, however, we observed that DTE was not well tolerated and could induce cell toxicity. Accordingly, we switched to TCEP, another common reducing agent. DTE and TCEP, while both powerful to break disulfide bonds, differ in their membrane permeability (DTE can cross membranes but not TCEP). This may explain the differential tolerability of HEK cells to these agents.

In order to ascertain that DTE and TCEP have similar effects on GluN1/GluN3A receptors and gain consistency between our oocyte and HEK experiments, we repeated the oocyte experiments presented in Figure 3E of the initial submission using TCEP instead of DTE. As illustrated in the new Figure 3, TCEP radically changes the shape and amplitude of the GluN1/GluN3A receptor responses, exactly as observed with DTE. In the revised manuscript, we now show this TCEP experiment in the Main Figures (new Figure 3E) while moving the initial DTE experiment to the Supplementary Information section (Supplementary Figure 3). Quantification for both DTE and TCEP is also shown in the bar graph Supplementary Figure 3B.

REVIEWERS' COMMENTS:

Reviewer #1 (Remarks to the Author):

The authors have made a thorough response to the suggestions from the initial review. In particular, the new experiments on neurons are very important.

I have no new comments and recommend publication.

Reviewer #2 (Remarks to the Author):

This revised manuscript presents important new data establishing that CGP-78608 can be used to demonstrate functional expression of 'excitatory' GluN1/GluN3A heteromeric glycine receptors in native tissue (P8-P12 hippocampal pyramidal neurons), confirming the hypothesis put forwards in the original manuscript. This greatly increases the impact of the paper, and should open up new lines of investigation on this understudied NMDA receptor subtype.

In addition, via careful revision of the text, the authors now clearly acknowledge prior work from multiple groups, which reported complex responses for GluN1/GluN3A receptors in the presence of GluN1 antagonists.

In the revised manuscript, and their letter of rebuttal, the authors fully address all of the concerns raised in my review of the original manuscript, and I recommend publication in Nature Communications with high enthusiasm.

Minor Comments

Lines 377-378, perhaps change to: serves both as an agonist and a functional antagonist

Line 400: into a long-lived glycine bound desensitized state

Mark Mayer

NCOMMS-18-14769A
Revised manuscript
Reply to reviewers

Reviewer #1 (Remarks to the Author):

The authors have made a thorough response to the suggestions from the initial review. In particular, the new experiments on neurons are very important.

I have no new comments and recommend publication.

Reviewer #2 (Remarks to the Author):

This revised manuscript presents important new data establishing that CGP-78608 can be used to demonstrate functional expression of 'excitatory' GluN1/GluN3A heteromeric glycine receptors in native tissue (P8-P12 hippocampal pyramidal neurons), confirming the hypothesis put forwards in the original manuscript. This greatly increases the impact of the paper, and should open up new lines of investigation on this understudied NMDA receptor subtype.

In addition, via careful revision of the text, the authors now clearly acknowledge prior work from multiple groups, which reported complex responses for GluN1/GluN3A receptors in the presence of GluN1 antagonists.

In the revised manuscript, and their letter of rebuttal, the authors fully address all of the concerns raised in my review of the original manuscript, and I recommend publication in Nature Communications with high enthusiasm.

Minor Comments

Lines 377-378, perhaps change to: serves both as an agonist and a functional antagonist

We have changed the text accordingly.

Line 400: into a long-lived glycine bound desensitized state

Idem.